# How Genetics and Genomics Advances Are Rewriting Pediatric Cancer Research and Clinical Care

**DOI:** 10.3390/medicina58101386

**Published:** 2022-10-02

**Authors:** Selene Cipri, Ludovico Abenavoli, Luigi Boccuto, Giada Del Baldo, Angela Mastronuzzi

**Affiliations:** 1Department of Hematology/Oncology, Cell Therapy, Gene Therapies and Hemopoietic Transplant, Bambino Gesù Children’s Hospital, 00165 Rome, Italy; 2Department of Health Sciences, University Magna Graecia, 88100 Catanzaro, Italy; 3Healthcare Genetics Program, School of Nursing, College of Behavioral, Social and Health Sciences, Clemson University, Clemson, SC 29634, USA

**Keywords:** pediatric cancer, Human Genome Project, next-generation sequencing, personalized medicine, cancer predisposition syndromes

## Abstract

In the last two decades, thanks to the data that have been obtained from the Human Genome Project and the development of next-generation sequencing (NGS) technologies, research in oncology has produced extremely important results in understanding the genomic landscape of pediatric cancers, which are the main cause of death during childhood. NGS has provided significant advances in medicine by detecting germline and somatic driver variants that determine the development and progression of many types of cancers, allowing a distinction between hereditary and non-hereditary cancers, characterizing resistance mechanisms that are also related to alterations of the epigenetic apparatus, and quantifying the mutational burden of tumor cells. A combined approach of next-generation technologies allows us to investigate the numerous molecular features of the cancer cell and the effects of the environment on it, discovering and following the path of personalized therapy to defeat an “ancient” disease that has had victories and defeats. In this paper, we provide an overview of the results that have been obtained in the last decade from genomic studies that were carried out on pediatric cancer and their contribution to the more accurate and faster diagnosis in the stratification of patients and the development of new precision therapies.

## 1. Introduction

There are an estimated 400,000 new diagnoses of cancer in children and adolescents (0–19 years), worldwide, every year [1,2]. According to the International Classification of Childhood Cancer (ICCC), the most common types of pediatric cancers include leukemias, lymphomas, and central nervous system cancers [3,4]. According to the Surveillance, Epidemiology, and End Results (SEER) more than 50% of pediatric cancers are rare, with there being an annual incidence of <200 cases [5].

In the pre-chemotherapy era, most cancers were treated with surgery and radiation and the cure rate of childhood cancer was <25% [6]. After the 1960s, with the introduction of chemotherapy and the advent of new diagnostic measures and the development of new therapies, a peremptory increase in survival was observed [6,7,8,9,10]. Over the last two decades, the data that have been obtained from the Human Genome Project and the development of next-generation sequencing technologies (NGS) with the advent of -omic sciences, in particular genomics, have clarified various molecular and genomic mechanisms of pediatric oncology, thus finding that the genomic landscape of cancer is very diverse and, in many cases, quite distinct from that of common adult cancers, e.g., in the etiology and tumorigenic mechanisms [6,7,11,12,13].

Studies that applied NGS technologies to cohorts of patients with pediatric cancer have highlighted the importance of having a genetic predisposition, as it has been estimated that the rates of pathogenic germline variants approach 10% and in some cases, such variants clearly contribute to the origin of the patient’s cancer, as in Li-Fraumeni syndrome; while in other cases, the contribution is ambiguous [14,15,16]. To date, over 100 cancer susceptibility genes have been described and most of the associated pathogenic germline variants are loss-of-function variants or variants that fall to the level of genes encoding DNA double-strand repair proteins [12,16,17,18,19].

The onset and progression of the tumorigenesis derives from a complex interaction between events affecting the germline and somatic events that rewrite the transcriptional landscape of the tumors, thus highlighting the importance of transcriptomics [20,21,22].

The cure rate of pediatric cancer is based on a chemotherapeutic and radiotherapeutic approach, which is often associated with side effects that can reduce the survivors’ quality of life [23]. In the process of investigating the molecular pathways that are involved in the development of pediatric cancer, new targets could be revealed for developing new therapies [6,7,15]. These latest findings would have a positive impact, in particular, on patients with relapsing/refractory tumors to conventional treatment, thus resulting in more effective and less toxic therapeutic protocols [7,24,25,26,27].

In this review, we discuss the results that have been obtained in the last decade from genomic studies that were carried out on pediatric cancer and their relevance from a clinical, diagnostic, prognostic, and therapeutic standpoint. The important amount of data that have been obtained in the field of genomics requires the integration of large-scale -omic data sets that are generated through an international professional network and a multidisciplinary approach that allows the sharing of individual professional skills to formulate multiaxial diagnostic evaluations and personalized therapeutic plans and to maximize the power of the data-driven approaches to advance pediatric cancer research (Figure 1).

## 2. Next-Generation Sequencing Promises a New Era in Pediatric Oncology: Impact on the Patient from Diagnosis to Prognosis

The advent of NGS technologies has had a positive impact on patient management as the mutational profiles of tumors play an important role in their diagnosis, risk stratification, and prognosis, particularly in central nervous system (CNS) tumors [28]. Hematological malignancies were the first conditions to be assigned a diagnostic value, a classification, and a risk stratification related to the characterized genomic alterations [29].

From a diagnostic and research aspect, the use of a sequential NGS approach is essential, e.g., the use of targeted NGS testing can be followed by a WES or WGS procedure in the case of a negative result occurring as the knowledge and analysis of tumor alterations can be pathognomonic, in particular, in patients with amplifications, point mutations, or gene fusions. This inevitably leads to changes in the patient’s prognosis and therapy [30,31,32,33,34,35,36,37].

Over the past decade, NGS studies have shown that the application of the combined approaches of these revolutionary technologies is needed to guide the discovery of new variants and driver genes in pediatric cancer [11,14,38,39,40,41,42,43,44,45].

### 2.1. Large-Scale Next-Generation Sequencing in Pediatric Cancer Patients

The first large-scale sequencing studies of pediatric tumors underline a low overall mutational burden and they have allowed the identification of new driver genes, although, for some high-risk and highly aggressive tumors, a gene or a driver pathway was not identifiable [38,39,40,41]. A Whole-Genome Sequencing (WGS) analysis of pediatric cancer patient cohorts found that germline variants accounted for approximately 10% of cancer causative variants, although the frequency varied by tumor type, one example being osteosarcomas and neuroblastomas which have a higher and lower frequency, respectively [14,15,16].

In patients with T-cell acute lymphoblastic leukemia (T-ALL), acute myeloid leukemia (AML), and Wilms tumor, WGS approaches have enabled the identification of subtype-specific driver genes and the interactions between germline and somatic variants in the cancer’s development [46,47,48].

WES has determined that there is a minimal frequency of pathogenetic variants (approximately three variants per tumor) in children that are under the age of 5 years with hepatoblastoma (HB) [49,50,51], while in the first case of hepatocellular carcinoma (HCC), a high mutational degree and the coexistence of pathogenic variants in *CTNNB1* (*Catenin Beta 1*) and *NFE2L2* (*NFE2 Like BZIP Transcription Factor 2*) were detected by a WES analysis [52]. A study in 35 cases of rhabdoid tumor (RT) patients identified that there was a very low mutational rate, with variants recurring only in *SMARCB1* (*SWI/SNF related, matrix-associated, actin-dependent regulator of chromatin, subfamily B, member 1*), which appeared to contribute to the tumorigenesis [53].

The NGS studies found that the frequency of fusion genes is higher than it had been previously thought to be, and that previously unidentified gene rearrangements could be playing a driver role [15,16]. RNA sequencing (RNA-seq) analyses reliably detected the fusions of the pathognomonic genes and provided information on the activity of the gene transcription [54]. Indeed, common gene fusion events are important because they can be pathognomonic for a specific diagnosis, for example, *EWS*-*FLI1* in Ewing’s sarcoma (EWS) and *PAX*-*FOXO1* in rhabdomyosarcoma (RMS) and can help to determine the optimal therapies. For example, CNS gliomas presenting the V600E point variant in *BRAF* (*B-Raf Proto-Oncogene, Serine/Threonine Kinase*) are more likely to respond to specific *BRAF* inhibitors, while gliomas with fusion genes involving *BRAF* are more likely to respond to drugs that inhibit the downstream Mitogen-Activated Protein Kinase 1 (*MAPK*) pathway [55]. RNA-seq, as previously mentioned, is a powerful tool for the detection of gene fusions; it has been used to detect the presence of the *NUTM1* (*NUT midline carcinoma family member 1*) rearrangement in B-ALL patients, which appears to be associated with a favorable prognosis [56]. Richarte-Filho et al. found by analyzing the tissues from 26 Ukrainian patients from Chernobyl with thyroid cancer that were 10 years old, that 22 tumors harbored fusion oncogenes arising mainly through intrachromosomal rearrangements and 23 of the oncogenic drivers that were identified in this cohort activated abnormally the *MAPK* pathway, including the two somatic rearrangements resulting in the fusion of the *ETV6* (*ETS Variant Transcription Factor 6*) gene with *NTRK1* (*Neurotrophic Receptor Tyrosine Kinase 1*), and the fusion of *AGK* (*Acylglycerol Kinase*) with *BRAF* [57]. For sarcomas and leukemias, in particular, fusion genes can occur between the oncogenes or genes that regulate the expression of the oncogenes, thus playing a central role in carcinogenesis. However, for other childhood cancers that are primarily characterized by structural variants, functional fusion genes are not produced. The mechanisms by which these recurrent structural changes have oncogenic effects have been identified for osteosarcoma (translocations limited to the first intron of *TP53*) and medulloblastoma (structural variants that juxtapose sequences of *GFI1* (*Growth Factor Independent 1 Transcriptional Repressor*) or *GFI1B* (*Growth Factor) Independent 1 Transcriptional Repressor B*) that are proximal to the enhancer elements [58,59]. LaHaye et al. demonstrated that the combination of fusion-calling pipeline algorithms and a practice-based filtration strategy which was applied to the data that were obtained from RNA-seq allowed them to quickly and accurately identify 67 relevant gene fusions in their study cohort with a diagnostic yield of 29.3%, which included *RBPMS-MET, BCAN-NTRK1,* and *TRIM22-BRAF* gene fusions [60].

### 2.2. Combined NGS Approach, a Strong Strategy Underlying the Synergy between Research and Diagnostics

The application of combined approaches of next-generation sequencing techniques (WGS, WES, and RNA-seq) in the study of pediatric oncology has allowed the formulation of the diagnostic framework—upstream—and the prognostic and therapeutic one for young patients—downstream—through the identification of genomic alterations that are related to the predisposition of the development of a specific tumor, the detection of variants of pharmacogenetic significance, and the discovery of new therapeutic targets [14,15,16,42,43,61,62,63,64]. Gröbner et al. performed a WGS in 547 samples and a WES in 414 pediatric cancer samples (24 different histotypes), thus identifying a low overall incidence of single nucleotide variants (SNVs), with the exception of high-grade gliomas with line variants, biallelic germline in DNA repair genes, *MSH6* (*MutS Homolog 6*) or *PMS2* (*PMS1 Homolog 2, Mismatch Repair System Component*); these rare tumors harbored more than 10 mutations per megabase [12].

Integrated studies of WGS and RNA-seq in ependymoma (PE) patients demonstrated the means by which these data can improve patient diagnosis and prognosis [65]. One study identified 11 ependymoma subgroups, including a subgroup that is characterized by a *c11orf95-RELA* fusion gene [66]. Instead, in another study on ependymomas, the regulatory super-enhancers of cancer-associated genes such as *PAX6* (*Paired* Box 6) were identified, and in studies on mice, it was observed that the inhibition of these super-enhancers in 60% of them was correlated with a positive impact on their survival [67]. The integration of WGS and RNA-seq contributes to the classification of tumors and also to the stratification that Is related to the risk of side effects, e.g., medulloblastoma [68,69]. Clarke et al. (2020) [70] performed a large study on a cohort of 241 cases of high-grade glioma (HGG) in children that were under the age of four. They used a variety of sequencing platforms including WGS, WES, and RNA-seq, thereby highlighting that high-grade childhood gliomas in the brain hemispheres comprise new subgroups, with a prevalence of *ALK* gene fusions *NTRK1/2/3*, *ROS1* (*ROS Proto-Oncogene 1, Receptor Tyrosine Kinase*) or *MET* (*MET proto-oncogene, receptor tyrosine kinase*) [70].

Overall, the discoveries that have been made employing next-generation technologies allow for the identification of new driver genes and carcinogenic pathways, also formulating a histotypic classification of pediatric cancers. Furthermore, these studies underline the value of the germline analysis method and the identification of variants in cancer predisposition genes.

## 3. Germline Variants and Cancer Predisposition Genes

The identification of the germline alterations in cancer predisposition genes is essential for understanding the etiology of the cancer, optimizing the initial therapy, and for developing surveillance protocols using cascade testing in families.

Several variants of the germline are characteristic of known predisposing syndromes such as *DICER1* (*Dicer 1, Ribonuclease III*) for pleuropulmonary blastoma (PPB), *RB1* for retinoblastoma (RB), *TP53* for adrenocortical carcinoma (ACC), and *SMARCB1* and *SMARCA4* for rhabdoid tumor and small cell ovarian cancer (SCCO), respectively [16]. The process of screening for germline variants of *SMARCB1* is recommended in children who are diagnosed with an atypical teratoid/rhabdoid tumor (AT/RT), as well as in relatives who may be unaffected carriers of it [71,72]. Rare cases of *SMARCB1* germline mosaicism in parents of affected patients have been described in the literature [71,73,74,75,76]. The germline variants of *SMARCB1* have been documented in patients with one or more primary brain and/or kidney tumors, according to them having a predisposition to the development of rhabdoid tumors [71,77]. Approximately one-third of patients with rhabdoid tumors have germline variants in *SMARCB1* [71,73]. In most of the cases, the variants are de novo and the mean age for the diagnosis of children with rhabdoid tumors that present with a variant or a germline deletion is younger (6 months) than that of children with an apparently sporadic disease (18 months) [72].

Paparella et al. identified a *POLR2A* (RNA Polymerase II Subunit A) germline variant in a patient with ependymoma. To date, ependymoma has never been reported in patients that are hosting pathogenic variants of *POLR2A*, consequently, this report opens the way for further studies to explore the possibility of a differential, clinical and functional impact of the different classes of these variants and their possible contribution to the predisposition to ependymomas [78].

The genomic studies of familial cases of Wilms tumor have allowed the identification of germline variants in the *DICER1*, *WT1* (*WT1 Transcription Factor*), *CHEK2* (*Checkpoint Kinase 2*), and *PALB2* (*Partner And Localizer Of BRCA2*) genes [46,79,80]. *TRIM28* (*Tripartite Motif Containing 28*) is a gene that is associated with an autosomal dominant form of inherited Wilms tumor predisposition, which accounts for approximately 8% of familial Wilms tumors and 2% of sporadic Wilms tumors. In a mutational study of 33 individuals, 21 of them had a variant in *TRIM28,* and all ten of the inherited variants were maternally transmitted [81]. A study was conducted to identify the variants in the *REST* (*RE1 Silencing Transcription Factor*) gene in nine out of five hundred and nineteen (1.7%) individuals with Wilms tumor and in their parents, who did not have a family history of overt disease, thus supporting the role of *REST* as a Wilms tumor predisposition gene [82].

Down syndrome (DS) is the most common cancer predisposition syndrome that has a higher risk of developing acute leukemia and a lower incidence of solid tumors. Boni et al. [83] identified a variant in *CTNNB1* that is associated with the *WNT* subgroup that is usually associated with a good prognosis in a patient with Down syndrome who presented with a medulloblastoma with focal aplasia and early metastatic recurrence [83]. In this study, Boni et al. highlight that the *NOTCH/WNT* dysregulation in DS—which is likely to be associated with an increased risk of leukemia—suggests that it has a pivotal role in the pathogenesis of MB; therefore, this condition should be further investigated in future studies using molecular characterizations [83].

Miele et al. carried out a study on 13 children that were diagnosed with adrenocortical tumors, a rare type of endocrine neoplasms. Twelve patients were disease free and one had a stable disease at the time of the study [84]. The genetic analyzes revealed that there were germline variants in *TP53* in 75% of the patients—five were inherited and one was de novo. One patient presented with Beckwith–Wiedemann syndrome, with a mosaic paternal uniparental disomy of chromosome 11 being detected in both the neoplastic and healthy adrenal tissue [84]. The results of the study demonstrated that there was an excellent prognosis, with an overall 5-year survival rate of 100% and a 5-year disease-free survival rate of 84.6%, and this supports the indication for genetic testing and family counseling for adrenocortical tumors [84].

An international genomic study of over 1200 patients with osteosarcoma found that there were pathogenic or possibly pathogenic germline variants in autosomal dominantly inherited cancer susceptibility genes in 18% of the patients. The frequency of these cancer susceptibility gene variants was higher in the children that were aged 10 years and younger [85].

A study of 150 children with solid tumors reported that there was a germline mutational prevalence of 10% [15]. A research group at the Memorial Sloan Kettering Cancer Center identified the pathogenetic or likely pathogenic variants in the cancer predisposition genes in 18% of the pediatric patients with solid tumors. These differences in mutational prevalence reflect differences in the patient cohorts, the genes that were analyzed, and the criteria for the inclusion of the variants affecting genes that are associated with autosomal recessive cancer predisposition syndromes [86].

Wang et al. analyzed the impact of the germline mutations in the cancer predisposition genes in long-term pediatric cancer survivors and reported that there was a prevalence of germline mutations of 5.8% in the study of 3006 survivors who were enrolled in the Saint Jude Lifetime Cohort Study (SJLIFE) [87,88]. Furthermore, the Childhood Cancer Survivor Study (CCSS) study identified *BRCA2* as a predisposition gene for pediatric or adolescent non-Hodgkin’s lymphoma [88].

In conclusion, these studies allow us to understand the importance of the knowledge of cancer predisposition and the awareness of the indications for the germline test based on the patient’s age and the phenotype of the patient, their family history, the presence of two or more primary tumors in the patient or the diagnosis of a rare tumor that is highly associated with a cancer predisposition gene.

## 4. Alterations in the Genes That Encode Proteins Involved in Epigenetic Regulation

Many diagnosed pediatric cancers are characterized by variants at the level of genes that code for epigenetic proteins which in turn regulate gene expression.

Several studies have been conducted on the proteins that are involved in epigenetic regulation, such as the SWI/SNF complex, which plays a role in chromatin remodeling. Alterations (gene fusions, deletions, etc.) at the level of the genes encoding this complex lead to the production of the non-functioning proteins that are identified in a multitude of pediatric and adult cancers, including synovial sarcoma, medulloblastoma, and kidney cancers [75,89].

A central component of this complex is encoded by the *SMARCB1* gene, and studies on patients with rhabdoid tumors have proven that they have a causative role for the pathogenic variants in this gene [53,90].

Wilms tumors have recurrent variants in their genes that are mostly involved in early renal development or epigenetic regulation (e.g., chromatin and miRNA modifications). Most of the cases of Wilms tumors with *AMER1* (*APC Membrane Recruitment Protein 1*) alterations have epigenetic abnormalities [91]. *AMER1* alterations are equally distributed between males and females, and *AMER1* inactivation has no apparent effect on the clinical presentation or prognosis [92]. Approximately 3% of children with Wilms tumors have epigenetic or genetic germline changes at the growth regulatory locus 11p15.5 without there being any clinical manifestation of an overgrowth. Like those children with Beckwith–Wiedemann syndrome (BWS), these children have a higher incidence of bilateral Wilms tumors or familial Wilms tumors [93].

Schwartzentruber et al. sequenced the exomes of 48 pediatric GBM samples and identified the mutations in the H3.3-ATRX-DAXX chromatin remodeling pathway in 44% of the tumours (21/48) [94]. Recurrent mutations in H3F3A, which encodes the replication-independent histone 3 variant H3.3, were observed in 31% of tumours, and this led to amino acid substitutions at two critical positions within the histone tail (K27M, G34R/G34V) that are involved in key regulatory post-translational modifications [94]. Similarly, diffuse intrinsic pontine gliomas are characterized by having frequent alterations in the gene encoding histone H3F3A [95].

About 20% of patients that are suffering from onco-hematological diseases are characterized by having variants in the genes that are encoding the CREB-binding protein, which is associated with alterations in histone acetylation [96].

Alterations in another gene that codes for proteins of the epigenetic apparatus, *SETD2*, are also common in all of the relapsed patients [97].

In more of 1000 pediatric cancer genomes, it was identified that there were *NSD2* (Nuclear Receptor Binding SET Domain Protein 2) histone methyltransferase alterations (p.E1099K), and in 14% of these, a t(12;21) *ETV6-RUNX1*-containing ALLs was detected [98]. Jianping et al. discovered that p.E1099K drives the mechanism of relapse in pediatric ALL [99]. In addition, most of the cases (up to 80%) of infant ALL and a few cases (5%) of childhood ALL are associated with the rearrangements of the *KMT2A* (Lysine Methyltransferase 2A) gene that encodes for a histone methyltransferase, which confers a poor prognosis [100,101,102,103].

In conclusion, the high frequency of the variants in genes that code for the regulators of the epigenetic apparatus suggests that there is a unique etiological function of pediatric cancers and also an important role in relapsing forms.

## 5. From the Origin of Carcinogenesis to Relapse: The Importance of the Mutational Signature

The analysis of the genetic fingerprint that is involved in the initiation of carcinogenesis and the understanding of its evolution has highlighted that there are, upstream, new scenarios in the relapsing cancer landscape, and downstream, the development of new preventive and therapeutic approaches.

In a study that was carried out on two cohorts of 26 adults and four pediatric patients with neuroblastomas, the COSMIC 18 signature that is associated with the exposure to reactive oxygen species (ROS) was identified [104]. Subsequently, an independent study found that COSMIC 18 manifests itself in the early stage of the carcinogenesis of neuroblastoma, and this signature is associated with a high expression of the mitochondrial genes of the ribosome and the electron transport chain, thus suggesting that there is a link between COSMIC 18 and ROS [105].

A signature that is associated with ultraviolet light exposure in melanoma has been found in a subgroup of patients with acute B-cell lymphoblastic leukemia [11,106,107]. Instead, an analysis of the patients with diffuse large B-cell lymphoma showed that 30% of them are under the age of 14, with them having a genetic signature that is similar to Burkitt’s lymphoma/leukemia [108,109].

A multitude of studies have focused on cisplatin therapy-induced signatures that are identified in patients with solid tumors, osteosarcoma, and brain tumors [106,110].

On the other hand, the most relevant analyses on the relapsing forms were carried out on patients who were suffering from onco-hematological diseases since the samples were easily available and therefore, more monitorable. Two concurrent mutational signatures in *CSF3R* (*Colony Stimulating Factor 3 Receptor*) and *CEBPA* (*CCAAT Enhancer Binding Protein Alpha*) that are related to higher relapse rates have been identified in patients with acute myeloid leukemia. In the latter, survival was not adversely affected by the administration of reduction therapy and stem cell transplantation [111].

Relapsing pediatric ALL are also characterized by variants in the *NT5C3* gene, which makes leukemia cells resistant to 6-mercaptopurine [112,113]. While the variants in the *CREBBP* gene of the relapsing patients confer that there is a resistance to glucocorticoids [114]. A study of relapsing pediatric ALL identified two new signatures, COSMIC 86 and 87, that are related to therapy; one was resulting from thiopurine treatment that was used during maintenance therapy. These were present in 27% of patients and accounted for 46% of the acquired resistance mutations in *TP53*, *NR3C1*, *PRPS1*, and *NT5C2* [115], which may explain the reported increased risk of relapse with intensified thiopurine maintenance [116].

Waanders et al. [117] published the results of a genomic analysis on the somatic alterations in a large cohort of relapsed pediatric ALLs which revealed that there are several mechanisms which are implicated in the hypermutations in relapsed ALL, including the alterations of MMR (mismatch repair) genes. Biallelic mutations in one of the MMR genes and high levels of single-base insertions or deletions in simple repeats may represent a mechanism of MMR-induced resistance to thiopurines in ALLs [117,118]. In addition, Fan et al [119] showed that in relapsed ALL-acquired *TP53* R248Q the mutations originate from the cooperation of a thiopurine treatment and an MMR deficiency and are associated with an increased risk of treatment failure [119].

Furthermore, therapy-induced variants are likely to induce secondary tumors in children, as demonstrated in a study in which patients with myoid malignancies that were treated with cisplatin and thiopurine had variants in the *TP53* and *RAS* pathways following their exposure to cytotoxic therapy. [120]. Finally, it has been shown that many patients have multiple subclones at the point of diagnosis, but that a single subclone can acquire additional mutations that confer resistance to therapy [121].

Antić et al. [122] assessed the clinical relevance and prognostic value of subclonal alterations in the relapse-associated genes *IKZF1, CREBBP, KRAS, NRAS, PTPN11, TP53, NT5C2*, and *WHSC1* in 503 ALL cases. The researchers identified 660 genomic alterations subclonal in 285 diagnosis samples. At the point of relapse, most of these subclonal mutations are lost, suggesting that their selective advantages over the wild-type clones during treatment is limited. In conclusion, for the genes that have been tested, there is no basis to consider subclonal alterations that are detected at the point of diagnosis as a prognostic marker [122]. In addition, in this study, the researchers indicate that the *RAS* pathway mutations were common, particularly in the minor subclones, and the comparisons between the RAS hotspot mutations revealed that there were differences in their capacity to drive clonal expansion in ALL [122].

Jerchel et al. [123] addressed the clinical value of the mutations in 13 key members of the RAS pathway in a cohort contained 461 newly diagnosed cases. This study showed that clonal mutations in *NRAS, KRAS, PTPN11,* and *FLT3* are associated with therapy resistance. Given that the clonal mutations at the initial stage of diagnosis were retained at the point of relapse and that the subclonal mutations often expanded at the point of relapse, RAS pathway mutations may serve as a biomarker to identify the patients that are eligible for MEK/ERK targeted therapy [123].

In conclusion, these data suggest the need to develop new therapeutic approaches in pediatric cancer patients to prevent the recurrence of secondary cancers.

## 6. Genomic Studies on Pediatric and Adult Cancers

The genomic studies of pediatric cancer that have been conducted highlighted that there are a variety of genetics differences between it and adult cancers. These studies identified the frequency of mutated genes and the formulation of the risk stratification and consequently, the most suitable therapies.

A whole exome sequencing (WES) on the DNA from bioptic samples in a cohort of patients with an average age of 10 years and who were affected by low-grade NF1 gliomas showed that they had a reduced number of mutations versus the adults [124].

Recent studies have shown that there is a high frequency of the variants in genes, such as *TP53* (Tumor Protein P53), *IDH1* [Isocitrate Dehydrogenase (NADP (+)) 1] and *ATRX* (Alpha Thalassemia/Mental Retardation Syndrome X-Linked) in younger patients with glioma, and that these variants have an age-dependent prognostic role, such as *ATRX* variants in patients with low-grade glioma who have shorter survival rate than adult patients do [125,126,127,128].

In pediatric solid tumors, an example of the stratification criteria is the subtype of the *WNT* pathway which is related to a good prognosis in children with medulloblastoma, who will receive less intensive care, consequently decreasing the short- and long-term side effects of the treatment [129,130,131]. Similarly, a stratification criteria for adult solid tumours is found in the colorectal subtypes: MSI-high subtypes are associated with favorable survival, whereas *BRAF*-mutated, *KRAS*-mutation-negative subtypes are associated with the highest rate of mortality [132].

Genomic studies of cancer are also important for developing new personalized therapies in pediatric cancer; an example of this is that the use targeted drugs for *ALK* alterations has shown excellent results in the treatment of anaplastic large cell lymphoma (ALCL) and childhood inflammatory myofibroblastic tumors [133].

Instead, for adults’ solid tumors, which are a larger and more widely studied cohorts, targeted therapy has been more frequently applied in their clinical management; an example of this is Imatinib, the first selective inhibitor of thyrosin-kinase to be approved for the treatment of leukemia. Today it is used as a neoadjuvant (preoperative) and adjuvant therapy (postoperative) for patients with gastrointestinal stromal tumors that are presenting mutations in proto-oncogene *KIT* [134].

Ma et al. [11] carried out a pan-cancer genome study and transcriptomic analysis on 1,699 pediatric patients with leukemia and solid tumors, and they found that the somatic alterations are mainly found at the level of *TP53, KRAS* (KRAS Proto-Oncogene, GTPase), *NRAS* (NRAS Proto-Oncogene, GTPase), *CDKN2A* (Cyclin Dependent Kinase Inhibitor 2A), or *NOTCH1* (Notch Receptor 1) [11]. They identified 142 probable driver genes, of which only 45% matched those that were found in the pan-oncology studies of adults [135,136,137,138], thus demonstrating that the copy number of alterations and the structural variants constitute the majority (62%) of the events [11]. Furthermore, among the neuroblastomas, at least one driver gene was identified in 72% of the tumors that were analyzed by WGS in comparison to there only being at least one driver gene in 26% of the samples that were analyzed by WES [11]. The WGS analysis also demonstrated the presence of chromothripsis (massive rearrangements that are caused by a single catastrophic event) in 11% of the samples [11]. De Rooij et al. performed a WES and RNA-seq on the samples from 99 patients (75 pediatric and 24 adults) to better understand the genomic landscape of non-Down syndrome acute megakaryoblastic leukemia (non-DS-AMKL) patients [139]. Their results demonstrated that pediatric non-DS-AMKL is a heterogeneous malignancy that can be divided into seven subgroups with varying outcomes. These subgroups are characterized by chimeric oncogenes with cooperating mutations in the kinase and epigenetic signaling genes. Taken together, these data shed light on the etiology of AMKL and provide useful information for treatment personalization practices [139].

Iacobucci et al. [140] compared the genomic characteristics of 159 pediatric and adult acute erythroleukemia (AEL) cases with non-AEL myeloid disorders and defined five age-related subgroups with distinct transcriptional profiles: adult, *TP53* mutated; *NPM1* mutated; *KMT2A* mutated/rearranged; adult, *DDX41* mutated; pediatric, *NUP98* rearranged. These genomic characteristics were proven to be able to influence the outcome: variants of NPM1 (Nucleophosmin 1) and the overexpression of *HOXB9* (Homeobox B9) are associated with a favorable prognosis, while the alterations of *TP53*, *FLT3* (Fms Related Receptor Tyrosine Kinase 3) or *RB1* (RB Transcriptional Corepressor 1) are associated with a poor survival rate [140]. In 45% of cases, there were recurrent variants of *ALK* (ALK Receptor Tyrosine Kinase) and *NTRK1* (Neurotrophic Receptor Tyrosine Kinase 1), that drive the erythroid leuke-mogenesis sensitive-to-tyrosine kinase (TRK) inhibition [140].

In the *KMT2A* rearrangement ALL subtype, a genomic alteration on the *RAS* pathway (commonly subclonal) is high prevalence in infants (about 90%) and low in adult (about 15%), and this is associated with a poor prognosis. ALL with *KMT2A* rearrangement have been shown to be sensitive to bortezomib or DOTIL inhibition [141]. In contrast, the *BCR-ABL1* t(9;22) (q34;q11.2) ALL subtype with *IKZF1* del and mut, *CDKN2A/B* del, has a prevalence of about 5% in children, while it is higher in adults (40–50%), and this is associated with a historically poor prognosis, which is improved with TKI [142,143,144].

In conclusion, the identification of genomic alterations has provided the opportunity to derive new targeted strategies for the clinical management of adult and pediatric cancers—referred to as precision medicine—including the non-invasive detection, risk assessment, molecular diagnosis, and targeted therapy of them.

## 7. The Road to Personalized Medicine

The genomic data that have been obtained in recent years in the field of oncology allow for a more accurate stratification of patients and the development of clinical trials for the formulation of increasingly personalized therapies [145].

An international prospective study of precision medicine, MAPPYACTS (NCT02613962), was aimed at defining the molecular profiles of the tumors in pediatric patients with recurrent/refractory malignancies in order to suggest the most suitable therapeutic protocol. The results of this study showed that 70% of patients had a genetic alteration that can be targeted for the development of a therapy. For 10% of these patients a drug that was “ready for routine use” already existed. Finally, this study highlights the need for new proof-of-concept clinical trials that address the molecular complexity of cancer [145].

In July 2016, the clinical trial called Next Generation Personalized Neuroblastoma Therapy (NEPENTHE) was initiated: it had been enrolling patients between 1-21 years old with a refractory or relapsed neuroblastoma with alterations of the *ALK* gene [146,147,148,149,150]. The patients that were enrolled in this trial were treated with a combination therapy of ribociclib, a third-generation cyclin-dependent kinase (CDK) inhibitor, and Ceritinib, an ALK inhibitor [151].

Further studies have employed computational approaches to identify the immunotherapy targets such as *CAMKV* in neuroblastoma patients presenting with *MYCN* or *GPC2* amplification in high-risk neuroblastoma [152,153].

Cacchione et al. [154] reported the clinical and radiological findings of a first-line treatment with everolimus, a selective mTOR inhibitor, in 10 patients that were diagnosed with mTOR-positive pediatric low-grade gliomas (pLGGs). The median duration of treatment was 19 months (range 13–60). Brain MRI showed that there was a stable disease in seven patients, a partial response in one of them, and a state of disease progression in two of them. The therapy-related adverse events were always reversible after a dose reduction or the temporary discontinuation of the treatment. As the authors suggest, these results provide preliminary support despite the low sample size that was used for the use of everolimus as a targeted therapy in pLGG showing a lack of progression with a manageable toxicity profile [154].

Lodi et al. [155] reported the case of a 13-year-old girl with Noonan syndrome (NS) that was associated with a recurrent variant in *PTPN11* who developed three different types of brain tumors, namely a glioma of the optic pathway, a glioneuronal neoplasm of the temporal lobe left, and a cerebellar pilocytic astrocytoma. The molecular characterization of the glioneuronal tumor allowed us to detect of high levels of phosphorylated mTOR (pmTOR); therefore, an everolimus-based therapeutic approach was chosen. The treatment was well tolerated and proved to be effective, thus leading to the stabilization of the tumor, which was surgically removed. The positive outcome of the present case suggests that practitioners should consider this approach for patients with RASopathies and brain tumors with hyperactivated *mTOR* pathways [155].

Several studies have highlighted the importance of immunotherapeutic approaches based on the neo-epitopes in pediatric cancer. Chang et al. [156] defined the neo-epitopic panorama of somatic alterations that were composed of missense variants and oncogene fusions in a study group of 540 genomes and transcriptomes of childhood cancer. As a whole, their results revealed that there was at least one neo-epitope in 88% of the leukemias, in 78% of the CNS, and in 90% of the solid tumors; 69.6% of the neo-epitopes were identified for the leukemias that were characterized by the *ETV6*-*RUNX1* fusion. These findings demonstrate that pediatric AMLs and ALLs require the further exploration of immunotherapy practices to identify the targets of leukemias that are characterized by fusion genes [115,156,157].

Proteins that are encoded by fused genes are an important class of drug targets, and they are important biomarkers for defining subgroups for risk stratification. To date, RNASeq—alone or in combination with WGS—has become a standard test for the research and clinical applications of new gene fusions [158,159,160,161].

In patients with Philadelphia chromosome-positive (Ph+) ALL, the tyrosine kinase inhibitors that are targeting the BCR-ABL1-fused protein are effective, e.g., imatinib mesylate [162]. In a study by the Children’s Oncology Group (COG), the combination of intensive chemotherapy and imatinib mesylate was administered daily, thus demonstrating a 5-year EFS rate that was 70% higher than the one that was related to imatinib mesylate alone [163,164].

In high-risk ALL patients, several target fusions have been identified, which involve a distinct number of kinases (e.g., *CSF1R*, *PDGFRB*, *ABL1*, or *ABL2*) and exhibit an ALL Ph-like signature, thus suggesting a therapeutic approach should be used that is based on therapy with tyrosine kinase inhibitors, as demonstrated in the Ph+ ALL patients [163,165,166].

Andolfo et al. [167] demonstrated that increased *EPHB4* (*EPH Receptor B4*) expression correlates with NB stage 4 and a poor overall survival. Furthermore, they revealed that the *EPHB4*-V871I gene variant correlates with an increased proliferating, migration, and invasion activity in two NB cell lines by targeting *VEGF*, *c-RAF,* and *CDK4* target genes and increasing ERK1 phosphorylation. Through the use of two *EPHB4* inhibitors, JI-101 and NVP-BHG712, the researchers were able to identify the phenotype that was driven by the variant. Overall, the results of the study suggested that *EPHB4* is a promising therapeutic target in high-risk NB [167].

Gene fusions have been detected in low-grade and high-grade non-cerebral glioma patients, in which gene fusions practices have involved the neurotrophin kinase receptor (*NTRK1*, *NTRK2*, and *NTRK3*) target [168,169,170].

The treatment of an HGG patient who was characterized by *ETV6-NTRK3* fusions with larotrectinib paved the way for the FDA approval of this pan-TRK inhibitor for the treatment of solid tumors with *NTRK* gene fusions [171,172].

Di Ruscio et al. [173] reported on a study of iHGG patients who received surgery and adjuvant chemotherapy; only two patients received the radiotherapy because their age at their diagnosis was over four-years-old. Molecular investigations, including next-generation sequencing (NGS) and DNA methylation, identified three gene fusions that involve *NTRK*, one *ROS1* fusion, one *MN1* rearrangement, and two *PATZ1* fusions. According to the molecular results, when the chemotherapy failed to control the disease, the two patients benefited from a target therapy with the *NTRK* inhibitor larotrectinib, thereby achieving complete remission, an excellent partial response, and no serious side effects [173].

Following the FDA approval of entrectinib [174] for the treatment of patients with solid tumors presenting with a *NTRK* gene fusion, the RNA-seq studies have identified the gene fusions involving *NTRK* in leukemias that exhibit a high sensitivity to TRK inhibition in mouse models, thus opening the door to new therapeutic opportunities with TRK inhibitors for leukemia patients who exhibit gene fusions involving *NTRK* [175,176].

In a genomic study of patients with juvenile myelomonocytic leukemia (JMML), 16 patients had no *RAS* variants, and in three of these that were 56 months of age or older, the researchers identified ALK/ROS1 tyrosine kinase fusions (*DCTN1-ALK*, *RANBP2-ALK*, and *TBL1XR1-ROS1)*. A patient with a fusion involving *ALK* was treated with crizotinib, an ALK, and ROS inhibitor, in addition to receiving conventional chemotherapy; a complete molecular remission was achieved [177].

The studies by Relling et al. [178] and Singh et al. [179] highlighted the importance of the analysis of variants in *TPMT* and *NUDT15* in cases of leukemia/lymphoma, given that some polymorphisms in these genes lead to the dysregulation of the metabolism of the therapeutic agents, thioguanine and mercaptopurine, thus suggest that knowing the *TPMT* polymorphisms and *NUDT15* is relevant at the start of therapy because allows for an appropriate chemotherapy dosage, thereby avoiding toxicity [178,179].

To date, scientific advances in the field of cellular and molecular biotechnology have led to a revolution in the field of personalized therapy in oncology, in particular in CAR-T (Chimeric Antigen Receptor T-cell) therapies [180,181]. Majzner et al. [182] presented the clinical experience of the first four patients with diffuse intrinsic pontine glioma (DIPG) or diffuse midline glioma (DMG) of the spinal cord, which is characterized by the H3K27M mutation, and they were treated with GD2-CAR T cells with a dose of 1 level equal to a 1 × 10^6^ cells/kg, which was administered intravenously. The patients who showed clinical benefit were eligible for subsequent infusions of GD2-CAR T cells which were administered intracerebroventricularly. Toxicity was largely related to tumor location, and it was reversible with supportive intensive care. The transcriptomic analyzes of 65,598 single cells from CAR T cell products and cerebrospinal fluid elucidated the heterogeneity of the response among the participants. These early results underscore the promise of this therapeutic approach for patients with DIPG or DMG in the spinal cord that is characterized by the H3K27M mutation [28,181,182,183].

Garcia Prieto et al. [184] recruited 114 patients with B-cell malignancies, including 77 patients with ALL and 37 patients with non-Hodgkin’s lymphoma, who were treated with CART19 cells (CAR T against CD19). The methylation analysis determined the epigenomic changes that occur in the patient’s T cells after the transduction of the CAR vector. The effects of the methylation status of different genomic sites were evaluated by their clinical response, cytokine release syndrome, neurotoxicity syndrome that is associated with immune effector cells, event-free survival, and overall survival, which were evaluated. In conclusion, the methylation status of the patient CART19 cells influences the efficacy of the therapy for patients with B cell neoplasia [184].

In conclusion, molecular investigations play a fundamental role in the diagnostic process and therapeutic strategy that is used. Their routine use in clinical practice could help to replace highly toxic chemotherapy regimens with target therapy that has moderate adverse effects, even in long-term follow-ups.

## 8. Conclusions

“We have two options: either to try to discover the genes important in malignancy by a piecemeal approach or to sequence the whole genome…” is reflection by Prof. Dulbecco that was reported in the article published in Science entitled “A Turning Point in Cancer Research: Sequencing the Human Genome”; this highlights the awareness of the complexity of cancer genetics [185,186].

With the application of NGS technologies in the field of pediatric oncology, a large amount of data have been obtained, leading to an exponential growth of the genomic databases that allow us to better understand the dynamics of the genomic architecture of cancer, the tolerability of its variation, and the degree of structural changes in the genome, thus deepening our understanding of the complexity of these conditions.

NGS approaches help to identify the phenotypic variability in several syndromes [187], and characterize different driver genomic alterations, thereby allowing for the stratification of the main pediatric cancer subtypes. Furthermore, these studies have made it possible to evaluate high-risk and relapsed/refractory cases, thus guiding and speeding up the design of new clinical trials in the formulation of increasingly personalized therapies.

Through the NGS studies on germline variants, the knowledge of cancer predisposition has broadened, thus allowing the development of surveillance protocols through cascade screening tests on family members, and highlighting the importance of oncological genetic counseling, both before and after the sequencing test.

Another relevant fact is the importance of genomic studies on patients with syndromes that are predisposed to the development of one or more tumors.

An integrative germline and somatic genomic approach to the diagnosis and relapse of cancer plays a central role in the treatment of pediatric cancer. The treatment strategies for children with germline alterations in cancer predisposition genes, particularly those involving chromosomal instability or DNA-repair loci, should minimize, where possible, the use of radiation or drugs that increase the risk of a second malignancy occurring.

However, there is a need to develop cheaper and faster tools for the characterization of variants, in particular for re-sequencing, as in the case of relapsing tumors, and reclassifications of variants of uncertain significance (VUSs). The data-driven approach to pediatric cancer research and the clinical treatments of it require the strength of an international multidisciplinary team to effectively translate these findings into new target therapies, overcome their limitations and improve the clinical outcomes of patients.

The growing relevance of genomics for clinical cancer care also highlights several considerable challenges, including the need to promote equal access to genomic testing that results in a health care revolution.

Finally, lifestyle choices, environmental exposures, microbiome, and cultural education must work in symbiosis with genomics to create a comprehensive view of the diversity of cancer biology.

## Figures and Tables

**Figure 1 medicina-58-01386-f001:**
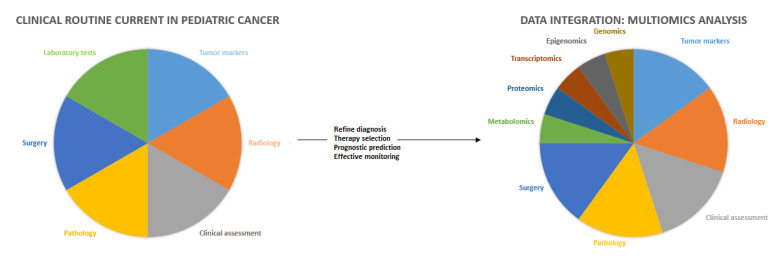
Clinical and genetic landscape of pediatric cancers: multidisciplinary network approach.

## Data Availability

Not applicable.

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
