# Peer review of "How Genetics and Genomics Advances Are Rewriting Pediatric Cancer Research and Clinical Care"

_medicina, 2022, doi:10.3390/medicina58101386_

Round 1

Reviewer 1 Report

This was a well written review on genetics in pediatric oncology.  The authors did a really nice job focusing on important points while also comprehensively summarizing the literature.

What is the main question addressed by the research? Is it relevant and

    interesting? This is a review article that evaluates the relevant publications in the last 10-20 years related to topics in the field of genetic testing for pediatrics.  It was a helpful summary of the current status of what is known in the field and was written in a clear and compelling way.

    How original is the topic? What does it add to the subject area compared

    with other published material? Pharmacogenomics is a growing field, but some barriers remain, making education about the topic an important step towards utilizing genetic testing more common.  A review article such as this provides a history and current status of the field that is helpful for those that require more knowledge in the area, but don’t have time to read through the varied literature on the topic.

    Is the paper well written? Is the text clear and easy to read? As stated above, I enjoyed reading the manuscript and felt that it was clear and understandable.

    Are the conclusions consistent with the evidence and arguments

    presented? Do they address the main question posed? While this was not a research article per se, the conclusions of the authors that a significant amount of genetic data is available for analysis and future studies is accurate.  With the improvement of NGS this will hopefully become less of a barrier and genetic testing will become cheaper and available to the general public around the world. Their mention of VUSs and other challenges is accurate, as well as their references to other non-genetic factors that certainly play a role in cancer predisposition and response to therapy.

Author Response

We would like to take this opportunity to thank you for taking the necessary time and effort to review our manuscript. We are grateful to you for providing your expertise in reviewing our manuscript, without which it would be impossible to maintain the high standards of this journal.

Finally, we sincerely appreciate all your valuable positive comments.

Reviewer 2 Report

In this review, Cipri et al, present recent advances in pediatric cancer research fueled by genetics and genomic techniques. Multiple studies have shed light into the mutational landscape of this group of heterogeneous tumors that differ substantially from adult tumors from the same origin.

The authors arrange an exhaustive compilation of studies and point towards how these studies can implement new routines in clinics or favor personalize treatments. The review is well written and offers a useful overview on the topic.

Main suggestions would be:

1.     The manuscript is organized in several sections. However, there is a kind of overlap between the type of studies enumerated in each section. Could the authors please revise the position of the studies and redistribute some of them to the proper section?

2.     Authors state in several parts of the manuscript about the differences between infant tumors and equivalent adult tumors. However, it would be more helpful to dedicate a specific section or a clear paragraph to state main differences between adult and pediatric tumors at the genomic and epigenetic level and maybe relate them to or speculate about other differences existing between those tumors (aggressiveness, therapy resistance, etc). Please also illustrate that analysis with some examples.

3.     The conclusions are very well exposed. However, there are not many comments about the limitations imposed by the fact that many of these pediatric tumors are rare diseases and this implies difficulties for large cohort studies required for obtaining reliable genomic information . In addition, it would be also desirable that the authors would bring up some efforts to establish databases with genomic/transcriptomic information on specific pediatric tumors

4.     Figure 1 is rather generic and not focused on pediatric tumors. Could authors also reshape it and confront how is the clinical routine today in pediatric cancers and which would be the ideal one following the multidisciplinary application of "omic" techniques?

5.     A second figure would be desirable highlighting the major outcomes from genomic studies and how they are different from adult tumors in terms of mutations and type of variants.

Author Response

  1. The manuscript is organized in several sections. However, there is a kind of overlap between the type of studies enumerated in each section. Could the authors please revise the position of the studies and redistribute some of them to the proper section?

Thank you for the suggestion, we have arranged the overlap parts.

  1. Authors state in several parts of the manuscript about the differences between infant tumors and equivalent adult tumors. However, it would be more helpful to dedicate a specific section or a clear paragraph to state main differences between adult and pediatric tumors at the genomic and epigenetic level and maybe relate them to or speculate about other differences existing between those tumors (aggressiveness, therapy resistance, etc). Please also illustrate that analysis with some examples.

Thank you for your suggestion, we have created a chapter entitled "Genomic studies on pediatric and adult cancers" - See section 6, pages 9-10.

  1. The conclusions are very well exposed. However, there are not many comments about the limitations imposed by the fact that many of these pediatric tumors are rare diseases and this implies difficulties for large cohort studies required for obtaining reliable genomic information. In addition, it would be also desirable that the authors would bring up some efforts to establish databases with genomic/transcriptomic information on specific pediatric tumors.

Thank you for your valuable suggestion, we have expanded the chapter of conclusions about this topic - See section 7, pages 10-11.

  1. Figure 1 is rather generic and not focused on pediatric tumors. Could authors also reshape it and confront how is the clinical routine today in pediatric cancers and which would be the ideal one following the multidisciplinary application of "omic" techniques?

Thank you for your suggestion. We adapted the figure to the confront between clinical routine today in pediatric cancers and which would be the ideal one following the multidisciplinary application of "omic" techniques - See page 2.

  1. A second figure would be desirable highlighting the major outcomes from genomic studies and how they are different from adult tumors in terms of mutations and type of variants.

We thank the reviewer for the valuable feedback. We appreciate the proposed perspective and agree on the relevance of the suggested figure 2. However, we believe that the magnitude of the comparison implied in such a figure goes beyond the purpose of this article: we aimed to discuss the changes induced by the Human Genome Project in the clinical practice of pediatric cancer, focusing rather on guidelines and methods and presenting actual genomic data only as examples. Moreover, the heterogeneity of pediatric and adult cohorts would require further discussion that, again, we consider beyond the scope of our review.

Reviewer 3 Report

In the manuscript entitled "How genetics and genomics advances are rewriting pediatric cancer research and clinical care", by Cipri S. and colleagues, authors give a comprehensive overview of the state of the art in genomics and genetics of pediatric cancers, with particular focus on the NGS techniques, as well as recent advances in the field. Overall, manuscript is well written and provides interesting overview for scientists and clinicians interested in the genetics of cancer.

Major comments:

1) Section 2, line 84 - it is not clear what authors mean by "gradual NGS approach". This term should be better explained or this sentence should be rewritten. 

2) Section 4, lines 318-322 - Authors should discuss KMT2A and WHSC1 genes in the context of ALL

3) In the section 5 authors should discuss recently described thio-dMMR mutational signature (PMID: 32793890 and PMID: 35122027). Another interesting aspect in this section is the clinical relevance of subclonal mutations (PMID: 33147938 and PMID: 28972594), or mutational signatures in individual leukemic clones (PMID: 33540666).

Minor comments:

Page 4, line 141 - ALL-B should be B-ALL

Authors should check that all gene names are written in italic

Author Response

Many thanks to your comments that will certainly improve the quality of our manuscript.

Major revisions:

  • Section 2, line 84 - it is not clear what authors mean by "gradual NGS approach". This term should be better explained or this sentence should be rewritten.

Thank you for your comment. To better explain this sentence, we have changed in this way: “From a diagnostic and research aspect, the use of a sequential gradual NGS ap-proach is essential – e.g. the use of a Targeted NGS testing can be followed by a WES or WGS, in case of a negative result – as knowledge and analysis of tumor alterations can be pathognomonic, in particular, in patients with amplification, point mutations, or gene fu-sions. This inevitably leads to changes in prognosis and therapy “ -  See lines 84-86.

       2) Section 4, lines 318-322 - Authors should discuss KMT2A and WHSC1 genes in the context of ALL

The authors agree with your opinion and discussed KMT2A and WHSC1 genes in the context of ALL in the manuscript – See lines 324-331.

  • In the section 5 authors should discuss recently described thio-dMMR mutational signature (PMID: 32793890 and PMID: 35122027). Another interesting aspect in this section is the clinical relevance of subclonal mutations (PMID: 33147938 and PMID: 28972594), or mutational signatures in individual leukemic clones (PMID: 33540666).

Thank you for your precious advice. We decided to add a description of thio-dMMR mutational signature and the clinical relevance of subclonal mutations by arguing the articles you've suggested us. See lines 371-379, 386-401.

Minor comments:

Page 4, line 141 - ALL-B should be B-ALL

Authors should check that all gene names are written in italic

Thank you for this minor revision that certainly improve the quality of the work.
